# Genomic Characterization of *Bacillus pumilus* Sonora, a Strain with Inhibitory Activity against *Vibrio parahaemolyticus*-AHPND and Probiotic Candidate for Shrimp Aquaculture

**DOI:** 10.3390/microorganisms12081623

**Published:** 2024-08-09

**Authors:** Karla A. Soto-Marfileño, Zinnia Judith Molina Garza, Ricardo Gomez Flores, Vida Mariel Molina-Garza, José C. Ibarra-Gámez, Bruno Gómez Gil, Lucio Galaviz-Silva

**Affiliations:** 1Facultad de Ciencias Biológicas, Universidad Autónoma de Nuevo León, Ave. Universidad S/N, Cd. Universitaria, San Nicolás de los Garza 66455, Nuevo León, Mexico; karla.soto.mrfl@gmail.com (K.A.S.-M.); rgomez60@hotmail.com (R.G.F.); vidamolina25@gmail.com (V.M.M.-G.); 2Instituto Tecnológico de Sonora, Departamento de Ciencias Agronómicas y Veterinarias, Ciudad Obregón 85000, Sonora, Mexico; jose.ibarra@itson.edu.mx; 3Mazatlán Unit, Research Center for Food and Development (CIAD), Ave Sábalo Cerritos S/N, Mazatlán 82112, Sinaloa, Mexico; bruno@ciad.mx

**Keywords:** *Bacillus pumilus*, Sonora strain, whole genome, probiotic, *Vibrio parahaemolyticus*-AHPND, antimicrobial peptides (AMP)

## Abstract

Acute hepatopancreatic necrosis disease, caused by *Vibrio parahaemolyticus* strains carrying the *pir*A and *pir*B toxin genes (VpAHPND), has been causing great economic losses in Asia and America in the shrimp farming industry. Numerous strains are resistant to antibiotics. However, supplementation with probiotic antagonists has become a more desirable treatment alternative. Fourteen strains of microorganisms were assessed for their potential to inhibit VpAHPND in vitro activity. The bacteria with the highest activity were challenged with VpAHPND-infected Pacific white shrimp *Litopenaeus vannamei.* Furthermore, the genomic characteristics of probiotic bacteria were explored by whole-genome sequencing. We identified the Sonora strain as *Bacillus pumilus*, which possesses positive proteolytic and cellulolytic activities that may improve shrimp nutrient uptake and digestion. Challenge trials showed a low cumulative mortality (11.1%). *B. pumilus* Son has a genome of 3,512,470 bp and 3734 coding sequences contained in 327 subsystems. Some of these genes are related to the biosynthesis of antimicrobial peptides (surfactins, fengycin, schizokinen, bacilibactin, and bacilysin), nitrogen and phosphorus metabolism, and stress response. Our in vitro and in vivo findings suggest that *B. pumilus* Sonora has potential as a functional probiotic.

## 1. Introduction

Aquaculture has emerged as an alternative strategy for food production with several advantages compared with capture fisheries since it reduces the need to catch additional wild fish to meet the increasing demand, thus contributing to preserving fish species stocks. Currently, mariculture farms provide half of the global aquatic animal production for human consumption [1]. In 2022, fisheries produced 185 million tons, of which approximately 51% originated from aquaculture activity [2]. In this industry, shrimp aquaculture is of the utmost importance to the global economy, especially for countries with low or moderate economic development, such as those belonging to Latin America and Asia [3]. In México, shrimp production has high economic value, with a revenue of approximately 876 million U.S. dollars and a production of 198,000 tons [2]. México ranks seventh in crustacean aquaculture and the states of Sonora, Sinaloa, and Nayarit are the main producers [4]. Nevertheless, the occurrence of diseases has led to losses of millions of U.S. dollars [5]. Among the main infectious agents are viruses, such as the Taura syndrome virus (TSV) and the white spot syndrome virus (WSSV), but bacterial species of the genus *Vibrio* cause a disease called vibriosis in most species and are considered opportunistic pathogens for shrimp. However, in 2013, atypical epizootics were reported for the first time in shrimp farms in northeastern México (states of Sonora, Sinaloa, and Nayarit), caused by pathogenic strains of *Vibrio parahaemolyticus*, the etiologic agent of acute hepatopancreatic necrosis disease (VpAHPND) [6]. This disease is characterized by a severely atrophied hepatopancreas in its acute stage, resulting in production losses of 80% and serious economic effects [7]. In the most affected countries by this disease, USD 23.6 billion financial losses have been estimated [8]. Most recently, outbreaks have arisen in the USA (2017), Korea (2018), and Japan (2020) and are assumed to have taken place in other regions of Asia and Latin America [9,10,11].

Antibiotics are commonly used to control various bacterial diseases in shrimp aquaculture. However, the excessive amount used in this field has exerted a strong selective pressure toward the development of antibiotic resistance, producing bacterial adaptation through the horizontal flow of resistance genes by mechanisms of chromosomal mutation or plasmid acquisition. In addition, the increase in the use of antibiotics has led to their accumulation in shrimp meat, making it very difficult to introduce them into the international market because they may pose potential risks to consumer health [12]. Hence, an alternative strategy for the treatment of infections caused by pathogenic microorganisms is supplementation with beneficial bacteria that produce active metabolites with bactericidal activity against pathogens, such as probiotics, which are live microorganisms and, if properly administered, confer benefits to the host [13]. In aquaculture, probiotics have been defined as microbial supplements that have beneficial effects on the host by modifying the host-associated or ambient microbial community, ensuring better feed utilization or enhancing the nutritional value of feed, and improving the host’s response to disease through various mechanisms, including competition for nutrients and/or the inhibition of pathogens derived from the synthesis of antimicrobial peptides (AMP) [14,15]. Bacteria of the genus *Bacillus* have been used as biological control agents to reduce the co-occurrence of vibriosis in shrimp farms [16], as they are part of the gills, skin, and intestinal tract microbiota of shrimp and it has been found that several species of this genus present antagonistic activity against various microorganisms, as in the case of *Bacillus subtilis BT23*, *741* [17,18] and two other strains with significant inhibition against Vp_AHPND_, *B. subtilis* (K3), and *B. velezensis* [19,20]. *Bacillus* species have also been found in marine sediment and are ingested naturally by marine animals, such as shrimp, which feed in or on marine sediment [15]. Therefore, this work aimed to assess the in vitro antagonistic effect of fourteen bacterial isolates against VpAHPND, select the best antagonist candidate by in vivo challenging experiments with the Pacific white shrimp *Litopenaeus vannamei*, and perform genomic characterization to identify genes that produce AMP-related genes in this strain and determine their potential probiotic role in shrimp aquaculture.

### Ethics Statement

Pacific white shrimps used in this experiment were healthy farm-grown crustaceans. All bioassays were conducted following the 3R policy (replacement, reduction, and refinement) according to guidelines of the Animal Ethics Committee for Laboratory Animals from Universidad Autónoma de Nuevo León, México.

## 2. Materials and Methods

### 2.1. Origin of Bacterial Strains

Samples of green seaweed, seawater, saline sediment, red mangrove, Chinese clam, and hermit crab were collected from the coast of Sonora (26°42′17″ N 110°34′54″ W) and transported to the laboratory at 4 °C in sealed plastic bags (Nasco, Fort Atkinson, WI, USA). Invertebrates were dissected to collect internal organs, which were deposited in sterile 2% NaCl (*w*/*v*) (BD Dibico; Estado de México, México), after which 5 g of tissues and seaweeds were individually pulverized, homogenized in a grinder under aseptic conditions, and serially diluted (10^−1^ to 10^−6^). Saline sediments and seawater (5 g) were serially diluted. Trypticase soy agar and marine agar, supplemented with 2% NaCl were used to culture the serial dilutions. Colonies of subcultures from isolates were obtained to produce axenic cultures. All cultures were incubated for 24 h at 28–32 °C [18,19,20].

Fourteen strains of microorganisms were selected for this study, whose origins and nomenclature are detailed in Table 1. In addition, the pathogenic strain VpAHPND-D11F of *Vibrio parahaemolyticus* (NCBI accession number MH091007.1) was isolated from moribund larvae (*L. vannamei*) in a shrimp farm in Tobari, Sonora, México, and identified by sequencing the 16S rDNA with universal primers, based on comparisons with the nucleotide Basic Local Alignment Search Tool (BLAST) program, as previously reported [21]. Strains were stored in the Laboratorio de Patología Molecular y Experimental of the Facultad de Ciencias Biológicas at Universidad Autónoma de Nuevo León, México.

### 2.2. Assessment of Antibacterial Activity Based on the Well Diffusion Test

Isolated bacteria were cultured in trypticase soy broth with 2% NaCl (TSB + N; Becton Dickinson, Sparks, MD, USA) for 24 h at 28 °C, after which plates of trypticase soy agar with 2% NaCl (TSA + N; Becton Dickinson) were inoculated with 100 µL of the pathogenic strain VpAHPND D11F at a density of 1.5 × 10^7^ CFU mL^−1^ and spread with sterilized swabs. Next, four 0.5 cm wells were aseptically punched at equal distances from each other on TSA plates, and 100 µL of isolates were added in triplicate to the wells (~1.5 × 10^7^ CFU mL^−1^). In the fourth well, we plated 100 μL of sterile TSB + N as a negative control [22,23]. Plates were then incubated for 24 h at 30 °C to determine the isolates antagonistic activity against VpAHPND, as assessed by measuring the diameter of the inhibition halos (mm) between the well and the bacterial lawn with the ImageJ software Version 1.54j [24,25].

### 2.3. Assessment of Antibacterial Activity Based on the Double Layer Assay

This inhibition test was performed as previously described [26], with some modifications. Firstly, each isolate was previously cultured with a sterile swab in TSA + N for 24 h at 30 °C, after which a 0.5 cm circular piece was cut with a hollow punch and deposited over the surface of a plate of Mueller-Hinton agar (Becton Dickinson) with 2% NaCl. After incubation for 24 h at 30 °C, an overlay of 10 mL of semisolid Mueller-Hinton agar inoculated with 100 μL of the pathogenic strain VpAHPND was poured on the first layer. The assay was performed in triplicate, and the antagonistic effect of the strains against the pathogenic bacteria was evaluated [27]. After 24 h of culture at 30 °C, the diameters of the inhibition halos were measured using the ImageJ software.

### 2.4. Hemolytic Activity

A sample of bacterial strains freshly grown in TSA + N was obtained with an inoculation loop and cultured by cross streaking on blood agar (Dibico, Cuautitlán Izcalli, México) with one milliliter of heparinized human blood, after which plates were incubated in triplicate for 24 h at 30 °C. We expected α-, β-, and γ-types of hemolysis to be recorded [28].

### 2.5. Extracellular Enzymatic Activity

Protease production was determined in TSA with 2.5% skim milk (29). To determine amylase activity, isolates were grown in Mueller-Hinton agar supplemented with 1.5% starch, and the degradation halo was revealed by adding Lugol solution to the plate surface after 24 h of incubation [28]. The cellulolytic activity was determined in minimal medium (Sigma-Aldrich Química, Naucalpan, México) supplemented with 2% carboxymethylcellulose (CMC; Sigma-Aldrich Química, Madrid, Spain) and visualized with Congo red dye (Sigma-Aldrich Química). The enzymatic/hydrolytic activity of the strains was observed from the formation of solubilization/degradation halos around colonies [29].

### 2.6. Biofilm Rupture Assay

Since VpAHPND induces moderate to strong biofilm formation, the potential of isolated strains to prevent the development or destroy preformed biofilms of the pathogenic strain was investigated [28]. An aliquot of 100 μL of VpAHPND D11F with OD_560_ = 0.02 (1.0 × 10 ^6^ CFU/mL) was transferred to flat-bottomed 96-well microtiter plates and incubated for 24 h (irreversible attachment phase) to 48 h (mature biofilm formation) at 30 °C, without agitation for biofilm development. In addition, cell-free supernatants of the isolates, freshly grown in TSB + N, were obtained by centrifugation at 2000 rpm for 10 min, followed by filtration through a 0.2 μm membrane filter, and 100 µL were placed in each well of microtiter plates and incubated for 24 h at 30 °C. As a negative control, VpAHPND without supernatant was used. The biofilm biomass was analyzed using a crystal violet staining assay, as previously described [30]. Optical densities (OD) were read at 595 nm in a microplate reader (Asys UVM-340 spectrophotometer; Biochrom Ltd., Cambridge, UK). Each isolate was tested in triplicate.

### 2.7. Test for In Vivo Antagonist Assessment in Bioassays

The Sonora strains were selected for the challenge assays because they produced the best in vitro antagonist and enzymatic activities. Pacific white shrimp were kindly provided by Megalarva de Sinaloa, S. de R.L., México.

#### 2.7.1. Diet Preparation

The Sonora strain was inoculated in 500 mL of TSB + N and incubated for 24 h at 30 °C with agitation at 150 rpm on a gyratory shaker (Max-Q 4450; Thermo Scientific, Madison, WI, USA), and the bacterial concentration was adjusted to 2.5 × 10^8^ CFU mL^−1^, using the McFarland scale. Next, the bacterial biomass was centrifuged at 5000 rpm for 10 min (Allegra 64R Centrifuge; Beckman Coulter, Brea, CA, USA), and the pellet was washed with PBS and resuspended in 300 mL of sterile saline solution. Next, fish oil was mixed with 500 g of commercial feed for juvenile shrimp (Rangen, Buhl, ID, USA), and the diet (saline solution with probiotics) was sprayed over it. We used a diet without probiotics as a negative control [31,32]. Treatments were evaluated in triplicate. Fifteen shrimps weighing 0.9 g were placed in aquariums containing 30 L of seawater (35 g L^−1^) with individual aeration pumps, lids, and feeders, and were acclimatized for one week before treatment administration. The diet was then administered for one week before the challenge test with VpAHPND. Every day, water in the aquariums was siphoned to eliminate feces and remains of food. We monitored temperature, dissolved oxygen concentration (DO), pH, and salinity with a waterproof portable meter (Hanna HI98194, Cd. Mexico, México). Sixty percent of the water was replaced weekly with fresh water, and the aquariums were maintained with constant aeration [33].

#### 2.7.2. Challenge Test

The VpAHPND D11F strain was used for bioassays and was inoculated in TSB + N to obtain a fresh culture. Next, three milliliters were added to a flask with 2.5 L of TSB + N and incubated at 160 rpm and 30 °C, until reaching an OD_600_ = 1 (10^7^ − 10^9^ CFU/mL), after which 140 mL of inoculum were added to each aquarium. Sterile TSB + N was added as the negative control and the positive control consisted of pathogenic bacteria [7]. After inoculum application, shrimps were monitored every hour, and lethargic, erratically swimming moribund shrimps were collected for real-time PCR (qPCR) analysis. Mortality data were recorded and reported as accumulated mortality [34].

#### 2.7.3. Diagnosis of VpAHPND by qPCR

Samples of the stomach, hepatopancreas, and intestine were preserved in 70% ethanol and processed for DNA extraction. The IQ Real AHPND/EMS kit (Gene Reach Biotechnology Corp., Taichung City, Taiwan) was used following the manufacturer’s instructions to detect the presence of copies of the *pir*A and *pir*B genes (toxin) and the AHPND plasmid (FAM probe) of *V. parahaemolyticus*, as well as shrimp genomic DNA (VIC probe). The 7500 Fast Real-Time PCR platform (Applied Biosystems, Foster City, CA, USA) was used with the 7500 Fast System version 1.4.2 software [35].

### 2.8. Whole Genome Sequencing of the Sonora Isolate

#### 2.8.1. Extraction of Genomic DNA from a Sonora Isolate

As the Sonora strain was the best probiotic candidate, according to the in vitro and in vivo tests, it was selected for genomic DNA analysis. Genomic DNA extraction was performed from a 24-h culture in TSB medium with 2% NaCl (*w*/*v*) using the Wizard genomic DNA purification kit (Promega, Madison, WI, USA), following the manufacturer’s protocol.

#### 2.8.2. Sample Tagging

The genomic DNA library preparation was performed using the Nextera XT Library Preparation Kit (Illumina, San Diego, CA, USA), following the supplier’s instructions. Whole genome sequencing was conducted on the Illumina MiSeq platform (Illumina, San Diego, CA, USA) following the MiSeq v.3 protocol (paired-end reads, 2 × 300 bp, and 300 cycles).

#### 2.8.3. De Novo Assembly, Genome Annotation, Phylogenetic Analysis, and Identification of Genes Involved in the Biosynthesis of AMP-Related Genes

Filtering of sequencing data was performed with FastQC and TrimGalore (v0.6.10) (https://github.com/FelixKrueger/TrimGalore, accessed on 7 August 2024). The quality of the reads was analyzed using FastQC, and adapters were clipped using TrimGalore. The resulting reads were then assembled using SPAdes v3.12.0 and the a5 Miseq pipeline [36]. The annotation of the genome was conducted with the RAST tool (Rapid Annotation using the Subsystem Technology), and the antiSMASH platform version 7.10 [37] was used to search for genes involved in the biosynthesis of AMP. The phylogenetic position of the Sonora genome among the reference and representative genomes listed in the NCBI database was identified using PATRIC (Pathosystems Resource Integration Center (https://www.patricbrc.org, accessed on 7 August 2024), which includes the type-strain of *B. pumilus* from NCBI. The phylogenetic tree was built and edited in iTOL (Interactive Tree of Life, https://itol.embl.de/, accessed on 7 August 2024) using the neighbor-joining method. The bootstrap values were calculated based on 1000 computer-generated trees.

### 2.9. Statistical Analysis

The results of the inhibition assays were analyzed using one-way ANOVA and Tukey’s multiple comparison test with a confidence interval of 95%. In addition, data from the shrimp challenge bioassay were analyzed using one-way ANOVA and Dunnett’s multiple comparisons test to determine the differences between treatments and survival, respectively [38,39], with the statistical software MINITAB 17 (Minitab Statistical Software, Minitab Inc., State College, PA, USA) and GraphPad Prism 9 (GraphPad Software Inc., San Diego, CA, USA).

## 3. Results

### 3.1. In Vitro Evaluation of the Antagonistic Activity of the Isolates against VpAHPND D11F

Seven out of 14 isolates presented zones of inhibition against the pathogen in the well diffusion test. The Sonora isolate significantly (*p* < 0.05) caused the highest inhibition activity (Figure 1), with an average inhibition diameter of 7.4 mm, higher than those of the remaining isolates, which showed an average value between 3.3 mm and 4.9 mm (Figure 2). In the double layer assay, the Sonora isolate also had a significantly (*p* < 0.05) larger inhibition diameter of 14.3 mm, followed by the J1, 43, and G2.1 strains, with mean inhibition diameters of 12.2, 11.1, and 11, respectively (Figure 3).

### 3.2. Hemolytic Activity and Enzymatic Activity

Seven isolates (H3M, H2, Sonora, E2.1, HA, 43, and G2.1) showed γ-hemolysis (Figure 4 and Table 2), whereas G3.2 and Y119 presented α-hemolysis. In addition, G10 and J1 showed a β-hemolysis profile. Regarding isolates 32a and G2, it was not possible to determine the type of hemolysis since these isolates did not grow on blood agar. Regarding the enzymatic activities of the isolates, G2, G3.2, Sonora, E2.1, G10, 32a, and 43 showed proteolytic activity; amylolytic activity was observed only in H3M and G2, and all the isolates except H3M and 32a showed cellulolytic activity. The enzymatic and antagonistic activity of the isolates led us to identify the best isolates for further in vivo challenge. The isolates Sonora, J1, 43, and J1 had the highest antagonistic potential against the pathogenic bacteria in vitro and exhibited the production of enzymes related to improving digestion or nutrient uptake, which are characteristics of interest for declaring a microorganism as a probiotic (Table 2).

### 3.3. Inhibition of Preformed Biofilms

Results showed that no significant differences were observed in the 24 h biofilm. After 48 h, the cell-free supernatant of the Sonora isolate had an antibiofilm effect (mature biofilm) since it inhibited or ruptured the biofilm of VpAHPND until it reached an OD of 0.15. However, the cell-free supernatant of the G2 and E2.1 isolates showed a weak potential to prevent the biofilm formation of VpAHPND (Figure 5).

### 3.4. Challenge Trial of the Sonora Strain against VpAHPND and qPCR Results

Diets supplemented with the Sonora strain showed significantly (*p* = 0.05) lower cumulative mortality, close to that of the negative control at 11.1%, at 14 d post-infection, and the positive control started with lethargic and moribund specimens at two days, reaching 90% cumulative mortality at 14 d (Figure 6). One-way ANOVA and Dunnett’s multiple comparison test showed significant differences among trials as compared with the control (*p* < 0.05). Regarding qPCR results, we did not find DNA copies of the *pir*A and *pir*B genes in the samples of stomach, hepatopancreas, and intestine of Sonora and negative control samples.

### 3.5. De Novo Assembly, Annotation, and Phylogenetic Analysis of the Sonora Isolate

Results from the genome characterization of the Sonora isolate are described in Table 3. After filtering with TrimGalore to remove sequencing adapter sequences and de novo *assembly* following the a5 pipeline, we obtained 37 contigs, a genome with a length of 3,512,470 bp, a GC content of 41.7%, an N_50_ of 264,060, and a mean coverage of 190. After their assembly, whole genome annotation was performed using Rapid Annotation Subsystem Technology (RAST v2.0). We predicted and identified 3734 protein-coding genes in 327 subsystems (Figure 7), of which seven genes were related to the biosynthesis of bacteriocins and ribosomal synthesized AMP (bacitracin), six genes were related to copper homeostasis, two genes were related to resistance to cobalt–zinc–cadmium, five genes were related to secondary metabolism (alkylpyrone synthase and auxin biosynthesis), and 10, 11, and 41 genes were related to nitrogen metabolism, phosphorus, and stress response, respectively. The genome of isolated Sonora contained 80 RNAs, 70 tRNAs, and 10 rRNAs. The bacteria did not have plasmids (Table 2). A comparison of the studied genome with other *Bacillus* genomes revealed that the Sonora strain belonged to *Bacillus pumilus* (Figure 7).

#### Genes Related to the Synthesis of AMP-Related Genes

Based on the antiSMASH version 7.1.0 analysis, the predicted AMP in the *B. pumilus* strain Sonora consisted of an NRPS (nonribosomal peptide synthetase) resembling the surfactin biosynthetic gene cluster (BGC), a beta-lactone with a BGC similar to fengycin, a hybrid NI-siderophore, a terpene cluster with a BGC similar to schizokinen, an NRP-metallopore similar to bacillibactin, a hybrid cluster with an NRPS, a type I paenilamycin-like PKS, and two lichenysin-like NRPSs. Another type of secondary metabolite was identified with a BGC similar to bacilysin, and a terpene, RRE-containing, beta-lactone, RiPP-like and type 3 PKS with an undetermined BGC. Table 4 and Figure 8 show more detailed information regarding the subsystems contained in the genome.

A phylogenetic tree was constructed to identify the most closely related strains to the Sonora isolate. The closest references and representative genomes were identified using Mash/MinHash. PGfams were selected from these genomes to determine the phylogenetic location of *B. pumilus* Sonora, protein sequences were aligned with MUSCLE, and nucleotides from each of those sequences were assigned to the protein alignment. Our results show that the *B. pumilus* Sonora strain is closely related to *B. pumilus* NJ-M2 and *B. pumilus* SARF-032315750.8 (Figure 9).

## 4. Discussion

Shrimp farming has significantly contributed to aquaculture production worldwide. However, this production is affected by various diseases [40]. One of the most relevant pathogens is VpAHPND, considered a constant threat in the shrimp industry [41], generating significant economic losses, estimated at approximately more than USD 1 billion per year [2]. Several alternatives have been suggested for the eradication or control of shrimp diseases with agents different from antibiotics due to VpAHPND strains resistance [42]. Among them, the application of probiotics has been preferred as a better alternative, since they confer antagonism against pathogens and help to develop the shrimp immune system, provide nutritional benefits, and assist the intestinal mucosal barrier, and are environmentally friendly and compatible with aquaculture [43].

In this study, bacterial isolates were analyzed to determine their antagonistic potential against VpAHPND and their enzymatic activity, and identify several genomic characteristics of the best candidate as a probiotic. Our results showed that the *B. pumilus* Sonora strain had the best profile among the candidates. Reports on the antagonistic activity of bacteria against different pathogenic strains of *Vibrio* sp. have demonstrated that the use of probiotics is the best alternative strategy for the control of pathogens in shrimp farms [44] due to the production of secondary metabolites that contribute to the antagonistic activity, such as aryl-polyene, polyketide synthase (PKS), or bacteriocins that, through different mechanisms such as bacterial competition, release these molecules and inhibit other bacteria [45].

Enzymatic assays for proteases, cellulases, amylases, and the hemolytic activity of the isolates were performed, and the Sonora strain presented cellulolytic and proteolytic activity. In addition to being an excellent antagonist against VpAHPND, this isolate had enzymes that may improve digestive activity by breaking down complex macromolecules, increasing digestion or nutrient absorption [46]. The presence of digestive enzymes in probiotic bacterial strains, such as proteases, amylases, and lipases, may contribute to the digestive process and the production of nutrients in shrimp, which is beneficial for growth, survival, and health in shrimp culture systems [47]. Sonora isolates presented γ hemolysis, and thus their use in fish and shrimp will not have a negative effect [48].

Furthermore, the *B. pumilus* Sonora strain, in addition to other isolates, was selected as the best inhibitor of VpAHPND biofilm formation. The strains used in this study have marine origins on the Pacific Ocean coasts. In particular, the *B. pumilus* Sonora strain was isolated from saline sediment, and this bacterium is a rich source of beneficial compounds because it produces new bioactive and anti-biofilm compounds that have not been isolated from terrestrial species. Nithya et al. [49], who evaluated the effects of supernatants of cultures of *B. pumilus* S6-15 on the formation of gram-positive and gram-negative biofilms, demonstrated that bacterial culture supernatants inhibited the initial attachment and biofilm formation and dispersed the mature biofilm.

During the AHPND challenge assays, our results demonstrated the effectiveness of the Sonora strain as a probiotic since it protected shrimps from the pathogenic effects of the toxins produced by the *pir*A and *pir*B genes, with a cumulative mortality of only 11.1%. In addition, our negative qPCR results detected that these genes indicated the bactericidal effect of the Sonora strain to eradicate the pathogenic VpAHPND from the stomach, hepatopancreas, and intestine. Other *B. pumilus* strains have been reported in challenge assays with a cumulative survival of 75.6%, although mixed with *B. subtilis* and/or *B. licheniformis* [50]. *Bacillus* isolates B2 and BT also showed good probiotic properties and were safe for hosts [51]. *B. pumilus* and three other species (*Pseudomonas putida*, *P. fluorescens*, and *Micrococcus luteus*) are included in commercial probiotics for aquaculture (Prowins Biotech Private Ltd., Hyderabad, India), but they have not been evaluated against VpAHPND, or specifically in shrimp aquaculture [52]. A few bacterial isolates, O-741, *B. subtilis* K3, *B. velezensis* [18,19,20], and others from saline sediment have also been reported as excellent probiotics to eliminate VpAHPND, including *B. safensis* 13L LOBSAL and *B. pumilus* 36R ATNSAL [33]. Furthermore, *Bacillus* species have been used frequently as probiotics for the treatment and/or prevention of diseases in plants, humans, and animals because they produced bacteriocins, trypsin, lipase, amylase, and AMP [33], similar to the results reported here for the Sonora strain.

Based on phylogenetic analyses, the Sonora isolate was identified as *B. pumilus*. Among the various species of probiotics discovered, species of *Bacillus* have the best probiotic properties because they produce antimicrobial substances with activity against many bacteria of interest in aquaculture [53,54]. In addition to their sporulation potential, this characteristic extended the period of effectiveness of the bacteria, giving them a double advantage in terms of survival in various environments compared with other probiotics [53]. Some of the mechanisms by which *Bacillus* species inhibit pathogens, in addition to the synthesis of antimicrobials, involve competition for adhesion sites, inhibition of virulence gene expression by quorum quenching, and the production of lytic enzymes [54,55,56]. *B. pumilis* Sonora shares the highest similarity with *B. pumilus* NJ-M2 1408.93 and *B. pumilus* SAFR-032 315750.8. The former was isolated from soil samples of different agricultural lands [57], unlike the bacteria in this study, which were obtained from saline sediment. However, NJ-M2 produces bacteriocins against different pathogenic bacteria, such as *Staphylococcus aureus*, *S. epidermidis*, *Enterococcus faecalis*, and *Streptococcus pneumoniae*. Regarding *B. pumilus* SAFR-032 [58], genome annotation showed that it contained 3715 protein-coding genes, 21 rRNAs, and 72 tRNAs, whereas *B. pumilus* Sonora contained 24 more coding genes than SAFR-032. However, this strain contained 11 ribosomal rRNAs and two additional tRNAs compared with those of the Sonora strain. The genome of *B. pumilus* 36R ATNSAL, isolated from saline sediments, had a genome size of 3,941,096 bp, with a GC content of 41.1%, assembled into 283 contigs, an N50 of 49.07 kbp, and a total of 3947 coding genes, 14 rRNAs, and 70 tRNAs [33], and compared with the genome of *B. pumilus* Sonora, the genome of this strain had a length of 3,512,470 bp, a GC content of 41.74%, an N50 of 26.4 kbp, a variation of 4 rRNAs, and 240 fewer coding sequences compared with 36R ATNSAL (Table 5).

The antiSMASH assay showed that *B. pumilus* Sonora shares a *Bacillus* surfactin BCG similar to that of *B. velezensis* FZB42. This is another bacterium that exerts an antagonistic effect against plant pathogens by producing various antimicrobial agents [59], including surfactin, which is an amphiphilic cyclic nonribosomal lipopeptide that consists of a peptide acyclic heptameric acid attached to a beta-hydroxy fatty acid chain and interacts with the cell membrane. Surfactins inhibit the growth of *Salmonella* sp. biofilm and inhibit the adhesion of *Listeria monocytogenes* and *Pseudomonas fluorescens* by more than 54% [60]. Alternatively, the genome contains a BGC similar to fengycin, a cyclic lipopeptide used as a fungicide and synthesized by *Bacillus subtilis* as an immune response against fungal infection, acting by damaging the cell membrane of the fungal target [61]. Another predicted BGC is similar to the schizokinen siderophore of the gene cluster found in *Leptolyngbya*. Siderophores are low molecular weight molecules that function as iron chelators; many microorganisms secrete them to collect ferric iron from the environment [62]. Schizokinen is a siderophore of the hydroxamate type that is a powerful Fe^3+^ chelator and antioxidant [63]. Despite these characteristics, this siderophore has not been studied in the context of aquaculture. A BGC similar to bacilysin was also predicted in the Sonora isolate, an algaecide compound that disrupts the cell wall of algae and cell membranes of organelles, which explains why it is used to suppress the growth of dangerous algal species, as well as for its anticyanobacterial activity that helps mitigate the dangerous effects of algal blooms [64]. *Bacillus velezensis* GY65 has a core of genes coding for the synthesis of bacilysin, defensins, macrolactin, and surfactin, giving the bacterium antagonistic potential against pathogenic bacteria [65] in addition to supporting the prevention and control of diseases in aquatic animals. The genome of *B. pumilus* Sonora also contains a BGC similar to that of bacillibacteria, as bacillibactins E and F are characterized as bacterial siderophores that contain nicotinic and benzoic acid moieties [66]. Bacillibactin is a potential AMP against multidrug-resistant pathogens such as *Staphylococcus aureus*, *Enterococcus faecalis*, *Pseudomonas aeruginosa*, and *Klebsiella pneumoniae*; thus, it may have antagonistic activity against various pathogenic *Vibrio* species [67]. *B. subtilis* BSXE-1601 and *B. subtilis* (K3) were previously reported as antagonists against several pathogenic species of *Vibrio*, including VpAHPND [19,68], because their genome harbored genes encoding AMP similar to *B. pumilus* Sonora (bacillibactin, fengycin, surfactin, and subtilosin A), which have antagonistic activity against pathogenic strains, particularly against VpAHPND [68]. Therefore, the metabolites responsible for this inhibition may be those mentioned above. In addition, this bacterium may have high potential to be used as a multifunctional biological agent in aquaculture against different microorganisms. To verify whether the biosynthetic activity of the antimicrobial compounds found in *B. pumilus* Sonora is specific to this bacterium, comparative genomics studies of this and other *B. pumilus* species are needed.

In conclusion, this study demonstrated that the isolated bacterial Sonora strain, subsequently identified as *B. pumilus*, has probiotic potential in shrimp due to its antagonistic activity against one of the most important pathogens in shrimp aquaculture, VpAHPND, which shows a high mortality rate and resistance to antimicrobials commonly used in shrimp farms. In addition, *B. pumilus* Sonora has additional desirable characteristics to be considered as a probiotic, such as the potential to produce cellulase and protease enzymes that help aquatic organisms with nutrient uptake and digestion. Furthermore, the study of genomic sequences provided insight into the possible mechanisms by which *B. pumilus* Sonora antagonizes VpAHPND and provides evidence supporting its antimicrobial potential in aquaculture.

## Figures and Tables

**Figure 1 microorganisms-12-01623-f001:**
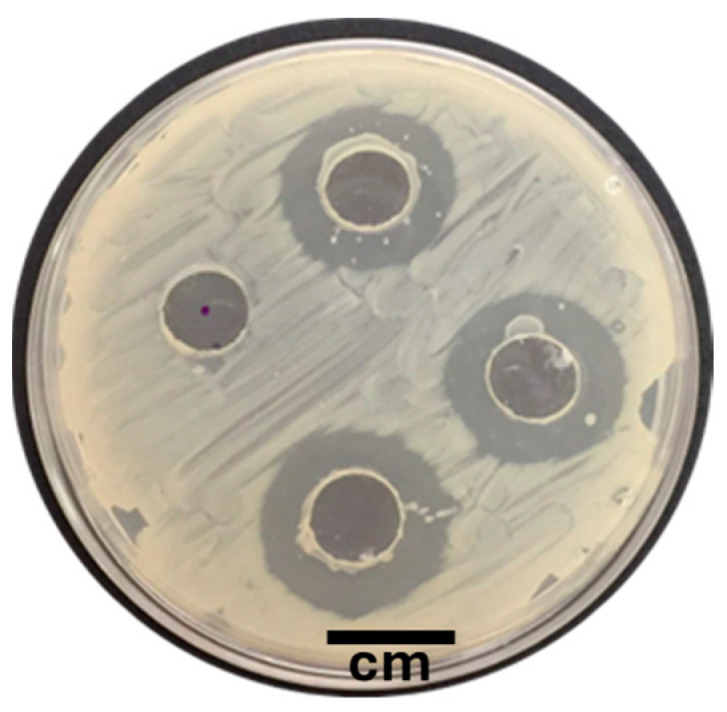
Antagonistic activity test of the isolate Sonora (Son) against *Vibrio parahaemolyticus* (VpAHPND) D11F. Analysis of inhibition zones produced by the well diffusion test.

**Figure 2 microorganisms-12-01623-f002:**
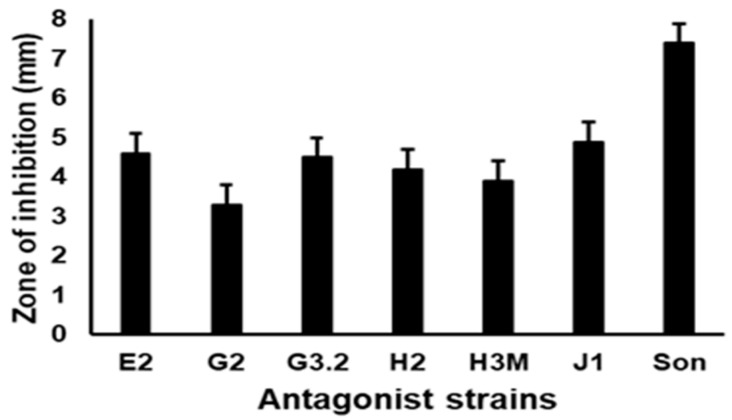
Inhibition halo (mm) of the antagonist strains vs. VpAHPND. Data represent the means ± SD of triplicate determinations. Columns with different letters indicate that the means were significantly (*p* < 0.05) different, according to Tukey’s HSD test.

**Figure 3 microorganisms-12-01623-f003:**
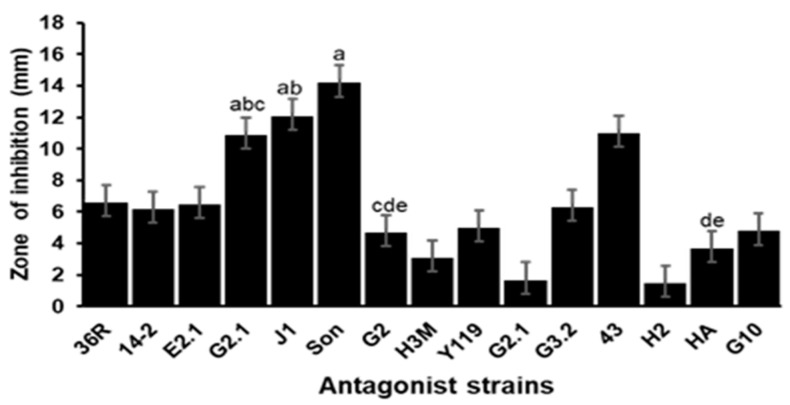
Double layer assay. Inhibition halo (mm) of antagonist strains against Vp AHPND D11F. Data represent the means ± SD of triplicate determinations. Columns with different letters indicate that the means were significantly (*p* < 0.05) different, according to Tukey’s HSD test.

**Figure 4 microorganisms-12-01623-f004:**
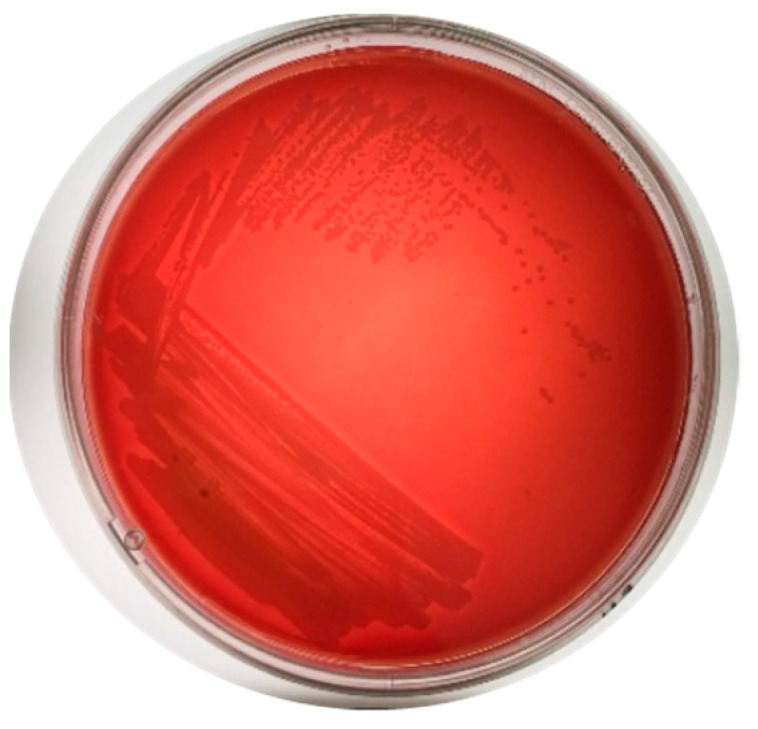
Hemolytic activity of the isolated Sonora against VPAHPND D11F. Representative γ hemolysis (lack of hemolysis) H3M, H2, and Sonora isolates via the well diffusion test.

**Figure 5 microorganisms-12-01623-f005:**
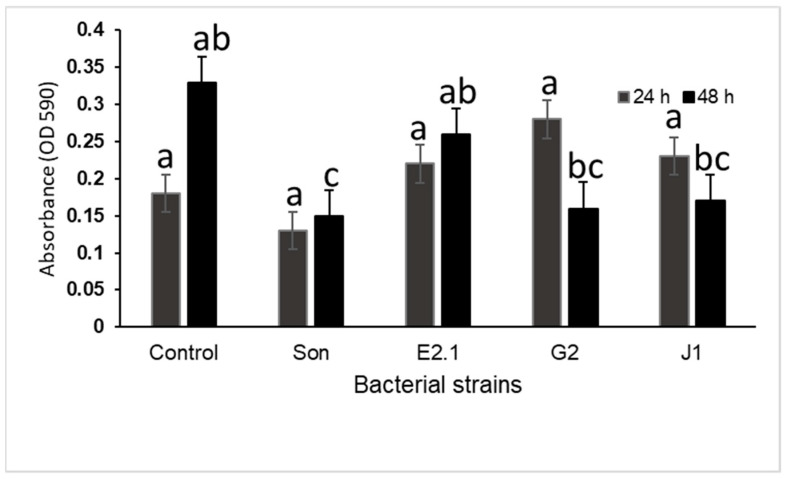
Inhibition of preformed biofilm. Data represent the means ± SD of triplicate determinations. ANOVA showed a significance level of *p* < 0.05. Columns with different letters indicate that the means were significantly (*p* < 0.05) different, according to Tukey’s HSD test.

**Figure 6 microorganisms-12-01623-f006:**
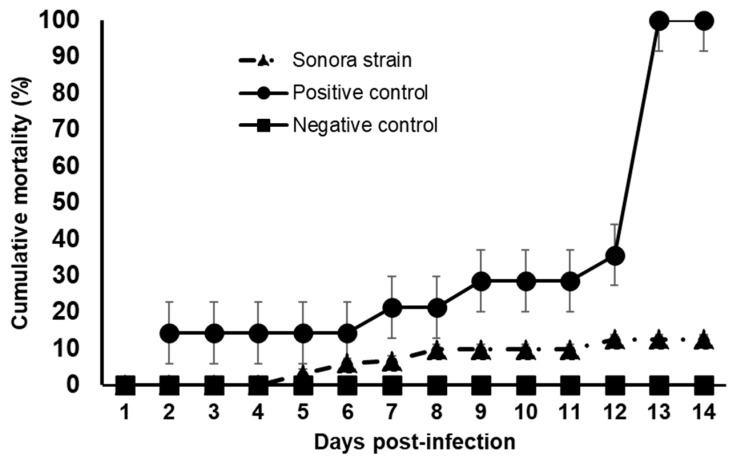
Cumulative mortality of the bioassays with the Sonora isolate. Data represent the means ± SD of triplicate determinations. The positive control and Sonora strain showed significant (*p* < 0.001) differences in Dunnett’s multiple comparisons test.

**Figure 7 microorganisms-12-01623-f007:**
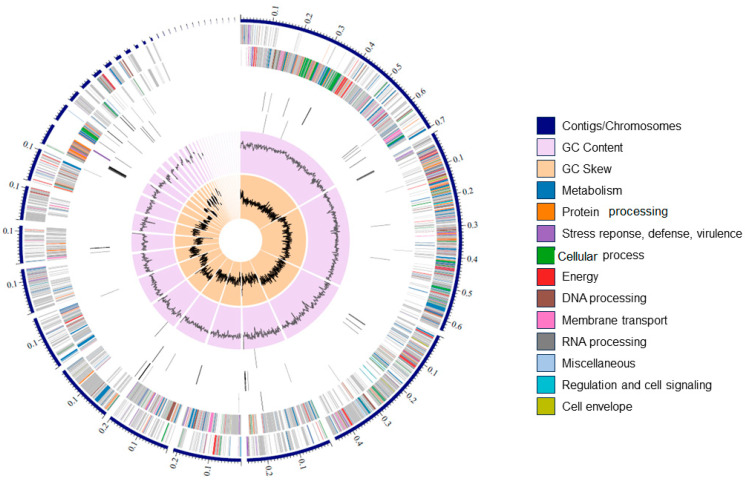
*Bacillus pumilus* Sonora genome. Circular graphic visualization of the distribution of genome annotations. External to internal rings, contigs, CDS forward strand (+), CDS reverse strand (−), RNA genes (rRNA and tRNA), and GC content.

**Figure 8 microorganisms-12-01623-f008:**
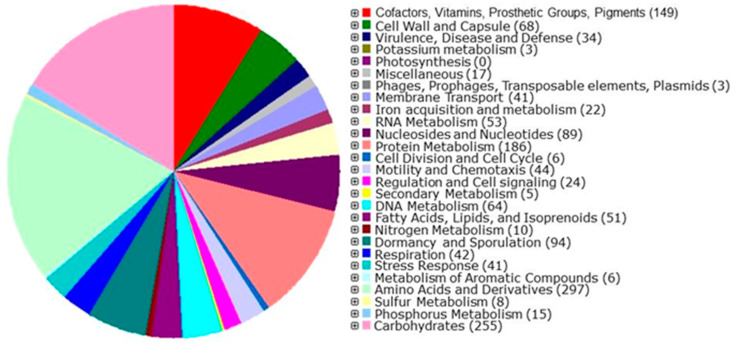
Schematic distribution of the categories of subsystems contained in the *B. pumilus* Sonora genome.

**Figure 9 microorganisms-12-01623-f009:**
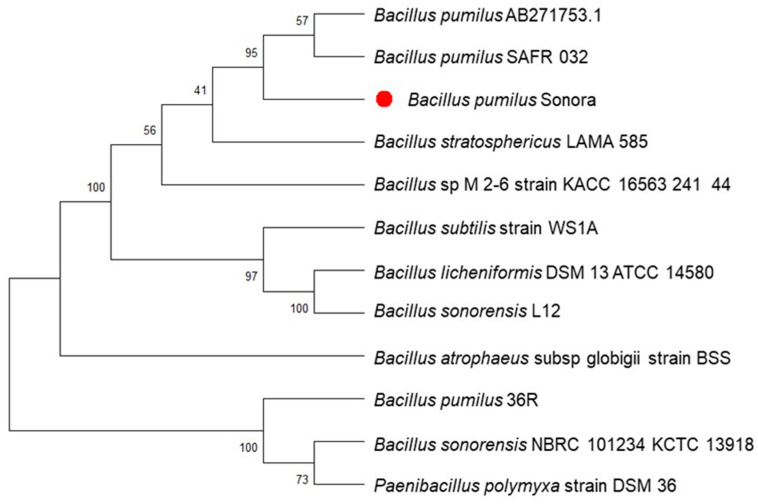
Phylogenetic tree of *Bacillus pumilus* Sonora (red circle) and its eleven related species based on 16S rDNA sequences. Multiple alignments of amino acids and nucleotides were concatenated into a data matrix via MUSCLE (v.3.8.1551). The maximum likelihood tree was constructed using RAxML v.8.2.12 software with 1000 bootstrap replicates to generate the support values in the tree.

**Table 1 microorganisms-12-01623-t001:** Nomenclature and source of isolates from the coast of Sonora, México.

Isolates	Sample Source
32a	Red mangrove (*Rhizophora mangle*)
43	Chinese clam (*Iliochione subrugosa*)
Sonora	Saline sediment
Y119	Seawater
HA	Mangle mud
J1	Seawater
H3M	Seawater
H2	Hermit crab (*Clibanarius panamensis*)
G2.1	Green seaweed (*Enteromorpha* sp.)
G2	Green seaweed (*Enteromorpha* sp.)
G3.2	Green seaweed (*Rhizoclonium* sp.)
E2.1	Seawater
G10	Saline sediment
36R	Saline sediment

**Table 2 microorganisms-12-01623-t002:** Enzymatic and hemolytic activity assays of the strains.

Strain	Hemolysis	Enzymatic Activity		
		Protease	Amylase	Cellulase
H3M	γ	−	+	−
H2	γ	−	−	+
Sonora	γ	+	−	+
E2.1	γ	+	−	+
HA	γ	−	−	+
43	γ	+	−	+
G2.1	γ	−	−	+
G3.2	α	+	−	+
Y119	α	−	−	+
G10	β	+	−	+
J1	β	−	−	+
G2	N/D	+	+	+
32a	N/D	+	−	−

Positive (+), negative (−), not determined (N/D).

**Table 3 microorganisms-12-01623-t003:** Genomic features of the *Bacillus pumilus* Sonora strain.

Feature	Counts
Genome length	3,512,470 bp
GC content	41.74%
Plasmid	0
Contig L_50_	4
Contig N_50_	264,060
Number of coding sequences	3734
Number of subsystems	327
tRNA	70
rRNA	10

**Table 4 microorganisms-12-01623-t004:** BCGs related to the synthesis of antimicrobials peptides (AMP) contained in the *Bacillus pumilus* Sonora genome.

Type	Most Similar Known Biosynthetic Gene Cluster
NRPS	Surfactin
Betalactone	Fengycin
Terpene	N/D
RRE-containing	Schizokinen
Betalactone	N/D
RiPP-like	N/D
T3PKS	N/D
NRP-metallopore	Bacillibactin
NRPS, T1KPKS	Paenilamicin
(2) NRPS	Lichenysin
Other	Bacilysin

**Table 5 microorganisms-12-01623-t005:** Comparison of genome characteristics of *B. pumilus* Sonora with *B. pumilus* SAFR 032 and 36R.

	*Bacillus pumilus* Sonora	*Bacillus pumilus* SAFR 032	*Bacillus pumilus* 36R
Genome length	3,600,890 bp	3,704,641 bp	3,941,096 bp
Contigs	37	1	283
%GC content	41.74%	41.5%	41.1%
N_50_	264,060 bp	3.7 Mb	49,075 bp
Protein-coding genes	3734	3710	3947
tRNA	70	72	70
rRNA	10	21	14

## Data Availability

The Whole Genome Shotgun project BioProject PRJNA980260 and BioSample SAMN35624284 for *B. pumilus* Sonora have been deposited at the DDBJ/ENA/GenBank under the accession number JASUAF000000000. Versions described in this manuscript correspond to JASUAF010000000 (https://www.ncbi.nlm.nih.gov/nuccore/JASUAF000000000, accessed on 24 July 2024).

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
