# Peer review of "Genomic Characterization of Bacillus pumilus Sonora, a Strain with Inhibitory Activity against Vibrio parahaemolyticus-AHPND and Probiotic Candidate for Shrimp Aquaculture"

_microorganisms, 2024, doi:10.3390/microorganisms12081623_

Round 1
Reviewer 1 Report
Comments and Suggestions for Authors
Review on “Genomic Characterization of Bacillus pumilus Sonora, a Strain with Inhibitory Activity Against Vibrio parahaemolyticus-AHPND and Probiotic Candidate for Shrimp Aquaculture” for manuscript ID microorganisms-3094245
In this manuscript the authors study genome features of Bacillus strain that has Vibrio pathogen biocontrol potential. The obtained data could help to improve our understanding about preventing shrimps’ diseases using probiotics.
In the brief introduction authors describe the prevalence and impact of Vibrio parahaemolyticus (AHPND) in shrimp farming and the need for effective control measures. Some bacteria such as Bacillus pumilus strain Sonora have potential in promoting shrimp health and preventing diseases. Unfortunately, some recent studies have been missed:
Proespraiwong P, Mavichak R, Imaizumi K, Hirono I, Unajak S. Evaluation of Bacillus spp. as Potent Probiotics with Reduction in AHPND-Related Mortality and Facilitating Growth Performance of Pacific White Shrimp (Litopenaeus vannamei) Farms. Microorganisms. 2023; 11(9):2176. https://doi.org/10.3390/microorganisms11092176
Chen Y-A, Chiu W-C, Wang T-Y, Wong H-c, Tang C-T (2024) Isolation and characterization of an antimicrobial Bacillus subtilis strain O-741 against Vibrio parahaemolyticus. PLoS ONE 19(4): e0299015. https://doi.org/10.1371/journal.pone.0299015
Chauyod, K., Rattanavarin, S., Sarapukdee, P., Porntheeraphat, S., Sritunyalucksana, K., & Khemthongcharoen, N. (2022). Bacillus velezensis suppression on the growth of Vibrio parahaemolyticus causing acute hepatopancreatic necrosis disease in marine shrimp. Journal of Applied Aquaculture, 35(4), 1202–1216. https://doi.org/10.1080/10454438.2022.2105672
Reference [2] might be replaced with more recent report https://doi.org/10.4060/cd0683en
Figure 2, 3: the bars’ patterns show no special information as each strain designated under the bar.
L425, Figure 9: the close species are poorly separated using 16S phylogeny only, the whole genome phylogeny could be more consistent.
L439-448: the comparison would be clearer as a table.
Methods section comments:
L104: the 16S sequence was released in 2018, how this Vibrio strain taxonomy was determined?
L232: I suggest to use the current version of AntiSMASH 7.1.0.
L223: sequencing statistics is required (e.g. number of reads)
The authors obtained genome data, but no raw sequencing reads were published or even mentioned to be published in future. Raw data need to be publicly available to reproduce authors’ results. The ready-to-use scenario (script) is highly affordable to add in the Github repository to produce the figures. BioProject PRJNA980260 contains the BioSamples only.
L513: The genome assembly NCBI accession is missing.
Minor corrections to the text:
L289, L397: what “g” stands for?
L416: missing bracket
L511: the “ symbol could be omitted
Author Response
Author´s Replay to the Review Report
Review on “Genomic Characterization of Bacillus pumilus Sonora, a Strain with Inhibitory Activity Against Vibrio parahaemolyticus-AHPND and Probiotic Candidate for Shrimp Aquaculture” for manuscript ID microorganisms-3094245
I would like to thank the reviewers for their careful and detailed critique of our manuscript. I have either replied to each specific comment addressed by them or have incorporated their suggestions in the revised manuscript. I hope that this revision will answer the thoughtful and significant concerns of the reviewers. (The crossed-out words were eliminated and replaced by those highlighted in yellow or red).
Reviewer 1
In this manuscript the authors study genome features of Bacillus strain that has Vibrio pathogen biocontrol potential. The obtained data could help to improve our understanding about preventing shrimps’ diseases using probiotics.
In the brief introduction authors describe the prevalence and impact of Vibrio parahaemolyticus (AHPND) in shrimp farming and the need for effective control measures. Some bacteria such as Bacillus pumilus strain Sonora have potential in promoting shrimp health and preventing diseases. Unfortunately, some recent studies have been missed:
Proespraiwong P, Mavichak R, Imaizumi K, Hirono I, Unajak S. Evaluation of Bacillus spp. as Potent Probiotics with Reduction in AHPND-Related Mortality and Facilitating Growth Performance of Pacific White Shrimp (Litopenaeus vannamei) Farms. Microorganisms. 2023; 11(9):2176. https://doi.org/10.3390/microorganisms11092176
Chen Y-A, Chiu W-C, Wang T-Y, Wong H-c, Tang C-T (2024) Isolation and characterization of an antimicrobial Bacillus subtilis strain O-741 against Vibrio parahaemolyticus. PLoS ONE 19(4): e0299015. https://doi.org/10.1371/journal.pone.0299015
Chauyod, K., Rattanavarin, S., Sarapukdee, P., Porntheeraphat, S., Sritunyalucksana, K., & Khemthongcharoen, N. (2022). Bacillus velezensis suppression on the growth of Vibrio parahaemolyticus causing acute hepatopancreatic necrosis disease in marine shrimp. Journal of Applied Aquaculture, 35(4), 1202–1216. https://doi.org/10.1080/10454438.2022.2105672
Response: We agree with this comment and have updated this information in the text, as follows “…as in the case of Bacillus subtilis BT23, 741 [17, 18] and two other strains with significant inhibition against VpAHPND, B. subtilis (K3), and B. velezensis [19, 20].
Reference [2] might be replaced with more recent report https://doi.org/10.4060/cd0683en
Response: Reference and information have been replaced as suggested “In 20220, fisheries produced 177.8 185 million tons, of which approximately 49 51% originated from aquaculture activity [2]…” and a production of 198,000 214,063 tons in 20221 [2]. Figure 2, 3: the bars’ patterns show no special information as each strain designated under the bar.
Response: Following your comment, the bar´s patterns were modified as follows:
Fig. 2
Fig. 3 L425, Figure 9: the close species are poorly separated using 16S phylogeny only, the whole genome phylogeny could be more consistent.
Response: In Materials and Methods section (lines 238 to 244 of the revised manuscript) a paragraph had been added: “The phylogenetic position of the Sonora genome among the reference and representative genomes listed in the NCBI database was identified using PATRIC (Pathosystems Resource Integration Center (https://www.patricbrc.org), it includes the type-strain of B. pumilus from NCBI. The phylogenetic tree was built and edited in iTOL (Interactive Tree of Life, https://itol.embl.de/), using the neighbor-joining method. The bootstrap values were calculated based on 1000 computer-generated trees.
In Results sections, Figure 9 was replaced as follows:
Figure 9. Phylogenetic tree of Bacillus pumilus Sonora and its eleven related species based on the 16S rDNA sequences. Multiple alignment of amino acids and nucleotides were concatenated into a data matrix by MUSCLE (v.3.8.1551). The maximum likelihood tree was constructed using RAxML v.8.2.12 software with 1000 bootstrap replicates to generate the support values in the tree.
L439-448: the comparison would be clearer as a table. Response: Regarding your observation, a Table had been added:
Table 5. Comparison of genome characteristics of B. pumilus Sonora with B. pumilus SAFR 032 and 36R.
Bacillus pumilus Sonora
Bacillus pumilus SAFR 032
Bacillus pumilus 36R
Genome length
3,600,890 bp
3,704,641 bp
3,941,096 bp
Contigs
37
1
283
%GC content
41.74%
41.5%
41.1%
N50
264,060 bp
3.7 Mb
49,075 bp
Protein coding genes
3,734
3,710
3.947
tRNA
70
72
70
rRNA
10
21
14
Methods section comments:
L104: the 16S sequence was released in 2018, how this Vibrio strain taxonomy was determined?
Response: This issue has been clarified as follows (lines 106 to 110 of the revised manuscript): In addition, the pathogenic strain VpAHPND-D11F of Vibrio parahaemolyticus (NCBI accession number MH091007.1) was isolated from moribund larvae (L. vannamei) in a shrimp farm in Tobari, Sonora, Mexico, identified by sequencing the 16S rDNA with universal primers, based on comparisons with the nucleotide Basic Local Alignment Search Tool (BLAST) program, as previously reported [21]. L232: I suggest to use the current version of AntiSMASH 7.1.0.
Response: Analyses were performed again, and we did not find any difference with our results. Evidence is shown as follows:
L223: sequencing statistics is required (e.g. number of reads)
The authors obtained genome data, but no raw sequencing reads were published or even mentioned to be published in future. Raw data need to be publicly available to reproduce authors’ results. The ready-to-use scenario (script) is highly affordable to add in the Github repository to produce the figures. BioProject PRJNA980260 contains the BioSamples only.
L513: The genome assembly NCBI accession is missing.
Response: The accession sequences of genome data and raw sequences had been released on July 18th, 2024 by NCBI, although there is a delay between when a genome is released and when it appears on all of our public resources, a new note was added: Data Availability Statement: The Whole Genome Shotgun projects BioProject PRJNA980260 and BioSample SAMN35624284 for B. pumilus Sonora have been deposited at the DDBJ/ENA/GenBank under the accession number JASUAF000000000. Versions described in this manuscript correspond to JASUAF010000000 (https://www.ncbi.nlm.nih.gov/nuccore/JASUAF000000000).
Minor corrections to the text:
L289, L397: what “g” stands for?
Response: This letter means gamma hemolysis, and had been clarified as a Greek letter “”, as follows (line 298 of the revised manuscript):
Table 2. Enzymatic and hemolytic activity assays of the strains.
Table 2. Enzymatic and hemolytic activity assays of the strains.
Strain
Hemolysis
Enzymatic Activity
Protease
Amylase
Cellulase
H3M
-
+
-
H2
-
-
+
Sonora
+
-
+
E2.1
+
-
+
HA
-
-
+
43
+
-
+
G2.1
-
-
+
G3.2
+
-
+
Y119
-
-
+
G10
+
-
+
J1
-
-
+
G2
N/D
+
+
+
32a
N/D
+
-
-
Positive (+), Negative (-), Not determined (N/D)
L416: missing bracket
Response: Bracket has been added (line 428)
L511: the “ symbol could be omitted
Response: The symbol “ had been deleted (line 528)
Reviewer 2
Comments and Suggestions for Authors
The article is well written and the experiment well organized. In the case of this work, it is necessary to check the Materials and Methods and supplement the diameters of the agar wells and check the diameter of 0.5 mm, which seems to me extremely small for any experimental work. Although I am not enthusiastic about using the term probiotic for the researched bacteria (because that term itself requires meeting a lot of strict parameters), I am still of the opinion that it can be used. Additionally, Greek alphabet must be used for presentation of hemolysis.
I would like to thank the reviewers for their careful and detailed critique of our manuscript, Reviewer´s comments on sticky notes, had been addressed as follows:
1. What are the advantages?
Response:(Line 35) This phrase has been clarified as follows: “since it reduces the need to catch additional wild fish to meet the increasing demand, thus contributing to preserving fish stocks species.”
2. State well diameter. (Line 116) “0.5 mm seem to be very small diameter...please recheck.” (Line 127)
Response: We agree with your good observation, it was a mistake, we meant 0.5 cm, and all of them were corrected. “four wells of 0.5 cm”(Lines 121, 132)
3.Use greek symbol:
Response: Greek symbols were replaced in Lines 140, 275-277, 290 4. “declaring a microorganism a probiotic”
Response: This phrase was clarified: “declaring a microorganism as probiotic” (line 294 and 295 of the revised manuscript).

Reviewer 2 Report
Comments and Suggestions for Authors
The article is well written and the experiment well organized. In the case of this work, it is necessary to check the Materials and Methods and supplement the diameters of the agar wells and check the diameter of 0.5 mm, which seems to me extremely small for any experimental work. Although I am not enthusiastic about using the term probiotic for the researched bacteria (because that term itself requires meeting a lot of strict parameters), I am still of the opinion that it can be used. Additionally, Greek alphabet must be used for presentation of hemolysis.

Author Response
Author´s Replay to the Review Report
Review on “Genomic Characterization of Bacillus pumilus Sonora, a Strain with Inhibitory Activity Against Vibrio parahaemolyticus-AHPND and Probiotic Candidate for Shrimp Aquaculture” for manuscript ID microorganisms-3094245
I would like to thank the reviewers for their careful and detailed critique of our manuscript. I have either replied to each specific comment addressed by them or have incorporated their suggestions in the revised manuscript. I hope that this revision will answer the thoughtful and significant concerns of the reviewers. (The crossed-out words were eliminated and replaced by those highlighted in yellow or red).
Reviewer 1
In this manuscript the authors study genome features of Bacillus strain that has Vibrio pathogen biocontrol potential. The obtained data could help to improve our understanding about preventing shrimps’ diseases using probiotics.
In the brief introduction authors describe the prevalence and impact of Vibrio parahaemolyticus (AHPND) in shrimp farming and the need for effective control measures. Some bacteria such as Bacillus pumilus strain Sonora have potential in promoting shrimp health and preventing diseases. Unfortunately, some recent studies have been missed:
Proespraiwong P, Mavichak R, Imaizumi K, Hirono I, Unajak S. Evaluation of Bacillus spp. as Potent Probiotics with Reduction in AHPND-Related Mortality and Facilitating Growth Performance of Pacific White Shrimp (Litopenaeus vannamei) Farms. Microorganisms. 2023; 11(9):2176. https://doi.org/10.3390/microorganisms11092176
Chen Y-A, Chiu W-C, Wang T-Y, Wong H-c, Tang C-T (2024) Isolation and characterization of an antimicrobial Bacillus subtilis strain O-741 against Vibrio parahaemolyticus. PLoS ONE 19(4): e0299015. https://doi.org/10.1371/journal.pone.0299015
Chauyod, K., Rattanavarin, S., Sarapukdee, P., Porntheeraphat, S., Sritunyalucksana, K., & Khemthongcharoen, N. (2022). Bacillus velezensis suppression on the growth of Vibrio parahaemolyticus causing acute hepatopancreatic necrosis disease in marine shrimp. Journal of Applied Aquaculture, 35(4), 1202–1216. https://doi.org/10.1080/10454438.2022.2105672
Response: We agree with this comment and have updated this information in the text, as follows “…as in the case of Bacillus subtilis BT23, 741 [17, 18] and two other strains with significant inhibition against VpAHPND, B. subtilis (K3), and B. velezensis [19, 20].
Reference [2] might be replaced with more recent report https://doi.org/10.4060/cd0683en
Response: Reference and information have been replaced as suggested “In 20220, fisheries produced 177.8 185 million tons, of which approximately 49 51% originated from aquaculture activity [2]…” and a production of 198,000 214,063 tons in 20221 [2].
Figure 2, 3: the bars’ patterns show no special information as each strain designated under the bar.
Response: Following your comment, the bar´s patterns were modified as follows:
Fig. 2
Fig. 3
L425, Figure 9: the close species are poorly separated using 16S phylogeny only, the whole genome phylogeny could be more consistent.
Response: In Materials and Methods section (lines 238 to 244 of the revised manuscript) a paragraph had been added: “The phylogenetic position of the Sonora genome among the reference and representative genomes listed in the NCBI database was identified using PATRIC (Pathosystems Resource Integration Center (https://www.patricbrc.org), it includes the type-strain of B. pumilus from NCBI. The phylogenetic tree was built and edited in iTOL (Interactive Tree of Life, https://itol.embl.de/), using the neighbor-joining method. The bootstrap values were calculated based on 1000 computer-generated trees.
In Results sections, Figure 9 was replaced as follows:
Figure 9. Phylogenetic tree of Bacillus pumilus Sonora and its eleven related species based on the 16S rDNA sequences. Multiple alignment of amino acids and nucleotides were concatenated into a data matrix by MUSCLE (v.3.8.1551). The maximum likelihood tree was constructed using RAxML v.8.2.12 software with 1000 bootstrap replicates to generate the support values in the tree.
L439-448: the comparison would be clearer as a table.
Response: Regarding your observation, a Table had been added:
Table 5. Comparison of genome characteristics of B. pumilus Sonora with B. pumilus SAFR 032 and 36R.
|
|
Bacillus pumilus Sonora |
Bacillus pumilus SAFR 032 |
Bacillus pumilus 36R |
|
Genome length |
3,600,890 bp |
3,704,641 bp |
3,941,096 bp |
|
Contigs |
37 |
1 |
283 |
|
%GC content |
41.74% |
41.5% |
41.1% |
|
N50 |
264,060 bp |
3.7 Mb |
49,075 bp |
|
Protein coding genes |
3,734 |
3,710 |
3.947 |
|
tRNA |
70 |
72 |
70 |
|
rRNA |
10 |
21 |
14 |
Methods section comments:
L104: the 16S sequence was released in 2018, how this Vibrio strain taxonomy was determined?
Response: This issue has been clarified as follows (lines 106 to 110 of the revised manuscript): In addition, the pathogenic strain VpAHPND-D11F of Vibrio parahaemolyticus (NCBI accession number MH091007.1) was isolated from moribund larvae (L. vannamei) in a shrimp farm in Tobari, Sonora, Mexico, identified by sequencing the 16S rDNA with universal primers, based on comparisons with the nucleotide Basic Local Alignment Search Tool (BLAST) program, as previously reported [21].
L232: I suggest to use the current version of AntiSMASH 7.1.0.
Response: Analyses were performed again, and we did not find any difference with our results. Evidence is shown as follows:
L223: sequencing statistics is required (e.g. number of reads)
The authors obtained genome data, but no raw sequencing reads were published or even mentioned to be published in future. Raw data need to be publicly available to reproduce authors’ results. The ready-to-use scenario (script) is highly affordable to add in the Github repository to produce the figures. BioProject PRJNA980260 contains the BioSamples only.
L513: The genome assembly NCBI accession is missing.
Response: The accession sequences of genome data and raw sequences had been released on July 18th, 2024 by NCBI, although there is a delay between when a genome is released and when it appears on all of our public resources, a new note was added:
Data Availability Statement: The Whole Genome Shotgun projects BioProject PRJNA980260 and BioSample SAMN35624284 for B. pumilus Sonora have been deposited at the DDBJ/ENA/GenBank under the accession number JASUAF000000000. Versions described in this manuscript correspond to JASUAF010000000 (https://www.ncbi.nlm.nih.gov/nuccore/JASUAF000000000).
Minor corrections to the text:
L289, L397: what “g” stands for?
Response: This letter means gamma hemolysis, and had been clarified as a Greek letter “g”, as follows (line 298 of the revised manuscript):
Table 2. Enzymatic and hemolytic activity assays of the strains.
Table 2. Enzymatic and hemolytic activity assays of the strains.
|
Strain |
Hemolysis |
Enzymatic Activity |
||
|
Protease |
Amylase |
Cellulase |
||
|
H3M |
g |
- |
+ |
- |
|
H2 |
g |
- |
- |
+ |
|
Sonora |
g |
+ |
- |
+ |
|
E2.1 |
g |
+ |
- |
+ |
|
HA |
g |
- |
- |
+ |
|
43 |
g |
+ |
- |
+ |
|
G2.1 |
g |
- |
- |
+ |
|
G3.2 |
a |
+ |
- |
+ |
|
Y119 |
a |
- |
- |
+ |
|
G10 |
b |
+ |
- |
+ |
|
J1 |
b |
- |
- |
+ |
|
G2 |
N/D |
+ |
+ |
+ |
|
32a |
N/D |
+ |
- |
- |
Positive (+), Negative (-), Not determined (N/D)
L416: missing bracket
Response: Bracket has been added (line 428)
L511: the “ symbol could be omitted
Response: The symbol “ had been deleted (line 528)
Reviewer 2
Comments and Suggestions for Authors
The article is well written and the experiment well organized. In the case of this work, it is necessary to check the Materials and Methods and supplement the diameters of the agar wells and check the diameter of 0.5 mm, which seems to me extremely small for any experimental work. Although I am not enthusiastic about using the term probiotic for the researched bacteria (because that term itself requires meeting a lot of strict parameters), I am still of the opinion that it can be used. Additionally, Greek alphabet must be used for presentation of hemolysis.
I would like to thank the reviewers for their careful and detailed critique of our manuscript, Reviewer´s comments on sticky notes, had been addressed as follows:
- What are the advantages?
Response:(Line 35) This phrase has been clarified as follows: “since it reduces the need to catch additional wild fish to meet the increasing demand, thus contributing to preserving fish stocks species.”
- State well diameter. (Line 116) “0.5 mm seem to be very small diameter...please recheck.” (Line 127)
Response: We agree with your good observation, it was a mistake, we meant 0.5 cm, and all of them were corrected. “four wells of 0.5 cm”(Lines 121, 132)
3.Use greek symbol:
Response: Greek symbols were replaced in Lines 140, 275-277, 290
- “declaring a microorganism a probiotic”
Response: This phrase was clarified: “declaring a microorganism as probiotic” (line 294 and 295 of the revised manuscript).

Round 2
Reviewer 1 Report
Comments and Suggestions for Authors
I would like to thank the authors for the improving the manuscript, but some concerns remain to be addressed.
Please add the DOI for ref. [11] https://doi.org/10.4060/cb2119en
L327: Figure 6; L350, Figure 7; specify complete strain name "Sonora"
L394: the term "probiotic" defined as microorganism already
L398: ref. [43] devoted to humans, not to shrimps or even marine animals
L401, L410: Please specify complete specie name here
L418: the sentence is hard to understand, how the Sonora strain properties related with ref. [48]?
L449: the reference is required to support your point.
L519-523: please rephrase for clarity and avoid repetition
L538: "projects" → "project"
Author Response
Comments and Suggestions for Authors
I would like to thank the authors for the improving the manuscript, but some concerns remain to be addressed.
Please add the DOI for ref. [11]
Response: DOI has been added as follows: https://doi.org/10.4060/cb2119en
L327: Figure 6; L350, Figure 7; specify complete strain name "Sonora"
Response: Strain name “Son” had been completed as “Sonora”, (line 321) Fig. 6, and Fig 7 (line 344)
L394: the term "probiotic" defined as microorganism already
Response: “probiotic microorganism” has been modified as “probiotics” (Lines 185 and 386 of the revised manuscript).
L398: ref. [43] devoted to humans, not to shrimps or even marine animals
Response: It was a mistake; Reference 43 had been replaced by
Kumar, S.; Verma, A.K.; Singh, S.P.; Awasthi, A. Immunostimulants for shrimp aquaculture: paving pathway towards shrimp sustainability. Environ Sci Pollut Res Int. 2023, 30, 25325-25343. https://doi.org/10.1007/s11356-021-18433-y.
L401, L410: Please specify complete specie name here
Response: Specie name of the strain was provided, as suggested (Lines 392 and 410 of the revised manuscript).
L418: the sentence is hard to understand, how the Sonora strain properties related with ref. [48]?
Response: It was a mistake, we meant “Wei, C.; Luo, K.; Wang, M.; Li, Y.; Pan, M.; Xie, Y.; Qin, G.; Liu, Y.; Li, L.; Liu, Q.; Tian, X. (2022) Evaluation of potential probiotic properties of a strain of Lactobacillus plantarum for shrimp farming: from beneficial functions to safety assessment. Front. Microbiol. 2022, 13, 854131. https://doi.org/10.3389/fmicb.2022.854131” https://doi.org/10.3389/fmicb.2022.854131 (Lines 407 to 409 and reference 48 of the revised manuscript).
L449: the reference is required to support your point.
Response: References [54, 55] have been added.
L519-523: please rephrase for clarity and avoid repetition
Response: We have rephrased the sentences for clarity and to avoid repetition (Lines 494 to 499 and 510 to 512 of the revised manuscript).
L538: "projects" → "project"
Response: “s” letter was deleted.
